# Coinfection of Two *Rickettsia* Species in a Single Tick Species Provides New Insight into *Rickettsia-Rickettsia* and *Rickettsia-Vector* Interactions

Yu-Sheng Pan,[a] Xiao-Ming Cui,[a] Li-Feng Du,[a,b] Luo-Yuan Xia,[a,b] Chun-Hong Du,[c] Lesley Bell-Sakyi,[d] Ming-Zhu Zhang,[a] Dai-Yun Zhu,[a] Yi Dong,[c] Wei Wei,[a] Lin Zhao,[b] Yi Sun,[a] Qing-Yu Lv,[a] Run-Ze Ye,[a] Zhi-Hai He,[c] Qian Wang,[a] Liang-Jing Li,[a] Ming-Guo Yao,[c] Tao Xiong,[a] Jia-Fu Jiang,[a] Wu-Chun Cao,[a] ⓘ Na Jia[a]

aState Key Laboratory of Pathogen and Biosecurity, Beijing Institute of Microbiology and Epidemiology, Beijing, People's Republic of China
bInstitute of EcoHealth, School of Public Health, Shandong University, Jinan, Shandong, People's Republic of China
cYunnan Institute for Endemic Diseases Control and Prevention, Dali, Yunnan, People's Republic of China
dDepartment of Infection Biology and Microbiomes, Institute of Infection, Veterinary and Ecological Sciences, University of Liverpool, Liverpool, United Kingdom

Yu-Sheng Pan, Xiao-Ming Cui, Li-Feng Du, Luo-Yuan Xia, and Chun-Hong Du contributed equally to this article. Author order was determined according to their contribution.

**ABSTRACT** Rickettsiae are obligate intracellular bacteria that can cause life-threatening illnesses. There is an ongoing debate as to whether established infections by one *Rickettsia* species preclude the maintenance of the second species in ticks. Here, we identified two *Rickettsia* species in inoculum from *Haemaphysalis montgomeryi* ticks and subsequently obtained pure isolates of each species by plaque selection. The two isolates were classified as a transitional group and spotted fever group rickettsiae and named *Rickettsia hoogstraalii* str CS and *Rickettsia rhipicephalii* str EH, respectively. The coinfection of these two *Rickettsia* species was detected in 25.6% of individual field-collected *H. montgomeryi*. In cell culture infection models, *R. hoogstraalii* str CS overwhelmed *R. rhipicephalii* str EH with more obvious cytopathic effects, faster plaque formation, and increased cellular growth when cocultured, and *R. hoogstraalii* str CS seemed to polymerize actin tails differently from *R. rhipicephalii* str EH *in vitro*. This work provides a model to investigate the mechanisms of both *Rickettsia-Rickettsia* and *Rickettsia*-vector interactions.

**IMPORTANCE** The rickettsiae are a group of obligate intracellular Gram-negative bacteria that include human pathogens causing an array of clinical symptoms and even death. There is an important question in the field, that is whether one infection can block the superinfection of other rickettsiae. This work demonstrated the coinfection of two *Rickettsia* species in individual ticks and further highlighted that testing the rickettsial competitive exclusion hypothesis will undoubtedly be a promising area as methods for bioengineering and pathogen biocontrol become amenable for rickettsiae.

**KEYWORDS** rickettsiae, coinfection, *Haemaphysalis montgomeryi*, tick, interference, competition

The genus *Rickettsia* includes Gram-negative, obligately intracellular bacteria with great diversity in arthropods. The vertebrate-associated *Rickettsia* involves both an arthropod vector and a vertebrate host (1–3). In the past 30 years, with advanced molecular techniques and enhanced surveillance, the number of known species within the genus *Rickettsia* has increased by at least an order of magnitude (4). Based on the biological and genetic characteristics, rickettsiae are classified into four groups: the spotted fever group (SFG), typhus group (TG), transitional group (TRG), and ancestral group (AG) (1, 5). Tick-borne spotted fever group rickettsiae (SFGR) is globally distributed and can cause life-threatening illnesses, such as Rocky Mountain spotted fever and Mediterranean spotted fever, among many others (6).

Address correspondence to Wu-Chun Cao, caowc@bmi.ac.cn, Jia-Fu Jiang, jiangjf2008@gmail.com, or Na Jia, jiana79_41@hotmail.com.

The authors declare no conflict of interest.

The traditional postulate states that superinfection or maintenance of two or three *Rickettsia* spp. in one tick species, generally, including an endosymbiotic *Rickettsia*, was prevented by blocking the transmission of one species from generation to generation (7). The lack of transovarial transmission of superinfecting rickettsia has been reported in the mutual exclusion of *Rickettsia rickettsii* with other species in *Dermacentor* and *Amblyomma* spp. ticks (7–9). In addition, the resistance of the ovaries to coinfection of *Rickettsia montanensis* or *Rickettsia rhipicephali* was observed in *Dermacentor variabilis* (10). *Amblyomma americanum* infected with *Rickettsia amblyommatis* were less likely to acquire *Rickettsia parkeri* than uninfected ticks (11). *Rickettsia buchneri*, an endosymbiont of *Ixodes scapularis* can prevent other rickettsiae from colonizing the black-legged tick and/or being transmitted transovarially (12, 13). However, recently, molecular evidence of co-occurrence of two or three *Rickettsia* species in a single tick specimen has been identified in field surveys (14, 15). This further advanced the debate as to whether an established infection by a *Rickettsia* species in ticks would preclude infection and maintenance of a second species (16).

Here, we isolated two *Rickettsia* species by plaque selection from an inoculum comprising a pool of 12 nymphs, the progeny of a single female tick, after high-throughput sequencing provided evidence of more than one species. The existence of approximately 25% of individual field-collected ticks and successful transstadial transmission of infection by both provisional species prompted us to believe that this phenomenon is not just occasional. We then compared the cellular growth when separately cultured or cocultured and actin-based motility (ABM) *in vitro* of these two *Rickettsia* species. This work provided models to further unveil molecular and cellular mechanisms of coinfection and suggests that a deeper definition of all core proteins in rickettsial coinfection or competition in ticks will provide a better understanding of *Rickettsia*-*Rickettsia* and rickettsiae-tick interactions.

## RESULTS

**Discovery and isolation of two *Rickettsia* species from the progeny of a single tick specimen.** We collected 143 host-seeking unfed adult *Haemaphysalis montgomeryi* ticks and 52 blood-feeding *H. montgomeryi* from goats in Dali City, Yunnan Province of southwestern China, from 2018 to 2019. Three engorged female ticks laid eggs, which subsequently molted into larvae and developed into nymphs after blood-feeding in our laboratory. A pool of 12 nymphs, the progeny of one of the female ticks, was used to isolate rickettsiae by inoculation onto Vero 81 cells (Vero CCL-81 cell line). Thirty days postinoculation, we observed rickettsial bacilli using Giemsa staining (Fig. 1A) and then amplified the SFGR-specific *omp*A gene (605-bp) from DNA extracted from the infected Vero cells for confirmation. The deduced sequence had 98.2% identity to *Rickettsia massiliae*, implying isolation of an *R. massiliae*-like bacterium. We then released the *Rickettsia* from the cells by semipurification for genomic sequencing. We found sequencing reads and whole-genome assemblies could align to both *R. massiliae* and *R. hoogstraalii* reference genomes (Fig. S1A and B in Supplemental File 1), suggesting that the isolate might contain two distinct species. We distinguished the assembled sequences into *R. hoogstraalii*-like genome and *R. massiliae*-like genome based on sequence identity to *R. hoogstraalii* and *R. massiliae* genomes (Fig. S1B in Supplemental File 1). Interestingly, the completeness of these two genomes was 95.9% (*R. hoogstraalii*-like genome) and 96.7% (*R. massiliae*-like genome). Later, we built a phylogenetic tree that indicated these two genomes were closely relative to *R. massiliae* and *R. hoogstraalii*, respectively (Fig. S1C in Supplemental File 1). We subsequently designed two pairs of specific primers for amplifying the *omp*A gene of the two possible *Rickettsia* species by PCR (Table S1 in Supplemental File 1) and obtained target amplicons (Fig. S1D in Supplemental File 1), confirming a mixed population.

To obtain pure *Rickettsia* isolates from the mixture, we then tried to plaque-purify these two rickettsiae. The dilution ($10^{-4}$) was selected for initial inoculation of the mixed isolates on plates observed for 20 days of plaque formation. Plaques were selected from 6 to 20 days postinfection and were inoculated onto fresh Vero 81 cells

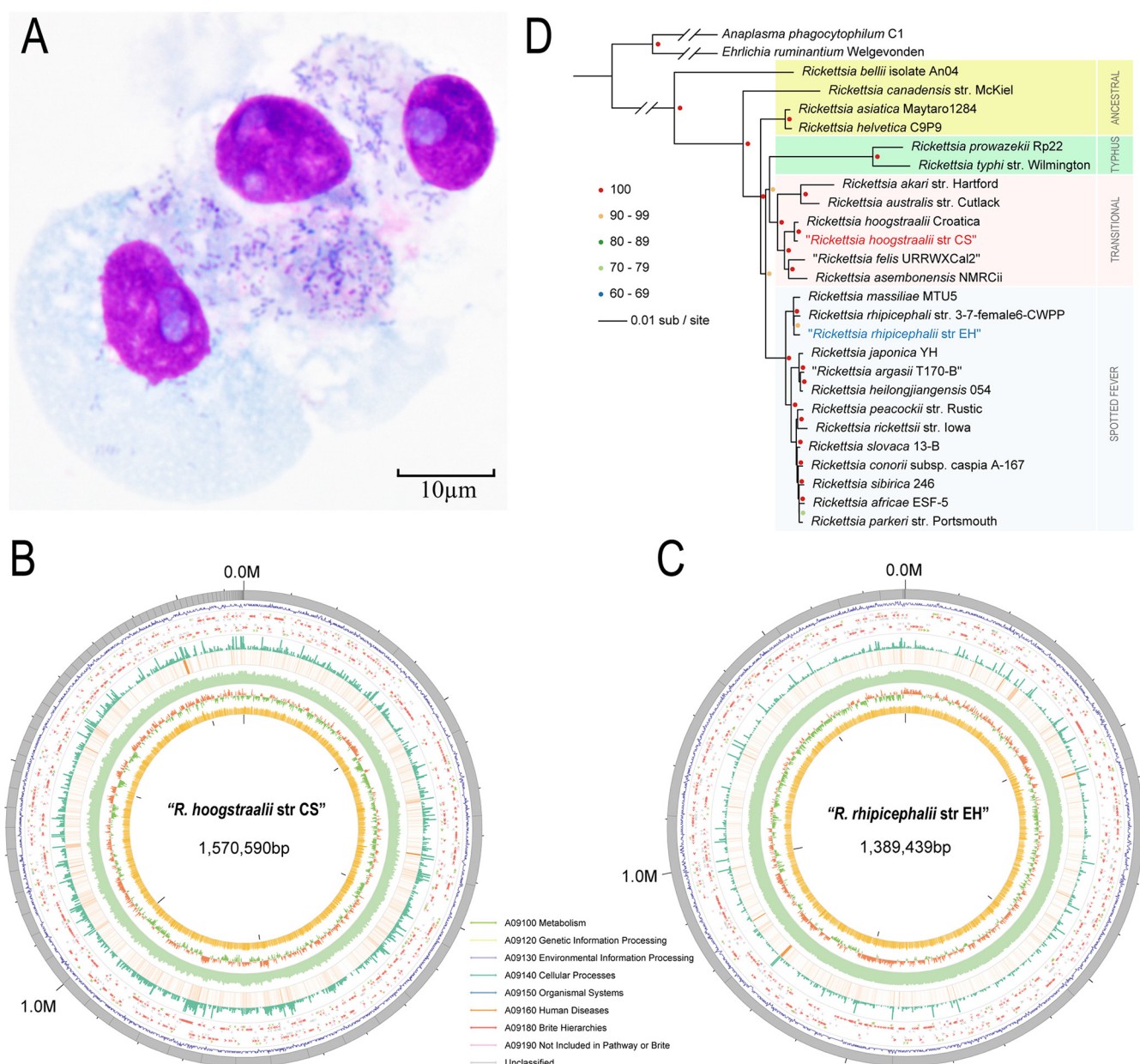

**FIG 1** Discovery and isolation of two *Rickettsia* species. (A) Giemsa staining of *Rickettsia* isolated from a pool of *Haemaphysalis montgomeryi* nymphs in Vero-81 cells. (B) Bird's eye view of the assembled genome of *Rickettsia hoogstraalii* str CS genome showing summary statistics. From the outer circle to the inner circle, eight types of information: contig length, genes density, gene annotation (arrow indicated direction color imply the KEGG of genes), TPM of genes, RNA sequencing data coverage, DNA sequencing data coverage, GC skew value, and GC content are labeled. (C) Bird's eye view of the assembled genome of *Rickettsia rhipicephalii* str EH genome showing summary statistics. Eight types of information are labeled in the same order as for *R. hoogstraalii* str CS. (D) The maximum likelihood phylogenomic tree of *Rickettsia rhipicephalii* str EH and *Rickettsia hoogstraalii* str CS was built with 25 other publicly available established or proposed Rickettsiales species. The tree was inferred by Raxml based on 332 single-copy orthologs identified by orthofinder. A total of 1,000 alternative runs were used to calculate support values. *Anaplasma phagocytophilum* and *Ehrlichia ruminantium* were two outgroup species to help root the tree. TRG and SFG were indicated with red and blue backgrounds, respectively.

for purification. Among a total of 89 plaques selected, we successfully purified two *Rickettsia* isolates (Fig. S2 in Supplemental File 1). The benchmarking universal single-copy orthologs (BUSCOs) analysis and comparisons to the most closely related genome of whole-genome sequencing of the two isolates supported their purification (Fig. 1B and C; Table 1; Fig. S1E in Supplemental File 1).

Based on whole-genome and important genes phylogenetic trees, two rickettsial isolates were close to *R. hoogstraalii* and *R. rhipicephalii* respectively (Fig. 1D, Fig. S3 in Supplemental File 1). *R. hoogstraalii* and *R. rhipicephali* belong to a transitional group

**TABLE 1** Genome characteristics of *Rickettsia hoogstraalii* str CS and *Rickettsia rhipicephalii* str EH

| Characteristic | *R. hoogstraalii* str CS | *Rickettsia rhipicephalii* str EH |
|---|---|---|
| Genome size (bp) | 1,570,590 | 1,389,439 |
| BUSCO | 99.70% (M, 0.3%; D, 0.0%) | 99.50% (M, 0.5%; D, 0.0%) |
| GC content | 32.75% | 32.49% |
| CDS[a] | 1602 | 1618 |
| tRNAs | 36 | 33 |
| No. of contigs | 78 | 14 |
| $N_{50}$ | 37,805 | 186,218 |
| N75 | 22,954 | 167,948 |
| % coding | 82.97% | 80.36% |
| rRNAs | 3 | 3 |

[a]CDS, coding sequence.

(TRG) rickettsia and SFG rickettsia respectively (17). According to the literature using core genome alignments to assign bacterial species (18), these two genomes are not novel species. We named these two rickettsiae *Rickettsia hoogstraalii* str CS and *Rickettsia rhipicephalii* str EH, respectively. *R. hoogstraalii* str CS as a TRG while *Rickettsia rhipicephalii* str EH as an SFG rickettsia were both detected and isolated in individual *H. montgomeryi* ticks.

The assembled genome size of *R. hoogstraalii* str CS, with an estimated length of 1,570,590 nucleotides (nt), is slightly larger than that of *R. rhipicephalii* str EH with an estimated length of 1,389,439 nt. The GC contents of the two rickettsiae were similar to each other and those of other *Rickettsia* spp. genomes, with 32.37% and 32.48%, respectively. The predicted numbers of open reading frames of *R. hoogstraalii* str CS and *R. rhipicephalii* str EH are 1602 and 1618, respectively. *R. hoogstraalii* str CS and *R. rhipicephalii* str EH also contain 36 and 33 tRNA-encoding genes, respectively, and 3 rRNA genes each (Table 1).

The two *Rickettsia* species had different plaque formation characteristics. *R. hoogstraalii* str CS- and *R. rhipicephalii* str EH-specific PCR on plaques selected between 6 to 10 days postinfection (dpi) showed almost all were *R. hoogstraalii* str CS, whereas, from 12 to 20 d.p.i, the plaques were larger, and the percentage formed by *R. rhipicephalii* str EH increased. Some plaques comprised a mixture of both *Rickettsia*, which could be due to the fusion of neighboring plaques (Fig. S2 in Supplemental File 1).

**Natural coinfection and transstadial transmission of the two *Rickettsia* species in *H. montgomeryi* collected from the field.** We examined a total of 195 *H. montgomeryi* adult ticks collected from Dali City by the specific PCR mentioned above and found the coinfection rate of *R. hoogstraalii* str CS and *R. rhipicephalii* str EH was 25.6% in individual ticks, which was significantly lower than the rate of single infection of *R. hoogstraalii* str CS (42.1%) but higher than that of single infection of *R. rhipicephalii* str EH (8.7%). The coinfection rate in engorged ticks (40.4%) detached from the goats in the field was significantly higher than the rate in unfed ticks (20.3%) (Fig. S4 in Supplemental File 1). The high coinfection rate in engorged ticks might simply mean there were greater copy numbers of both rickettsiae due to replication during the blood meal.

The nymphs remaining from the progeny of the individual female tick from which both *R. hoogstraalii* str CS and *R. rhipicephalii* str EH were isolated, were then used to evaluate transstadial transmission of the two bacteria. In the meantime, nymphal progeny from two other female ticks, one with only *R. hoogstraalii* str CS and one with only *R. rhipicephalii* str EH infection, were evaluated in parallel. A total of 10 nymphs from the mixed infection group were fully engorged and successfully molted to adult ticks. We tested three of these adults and found two ticks coinfected with *R. hoogstraalii* str CS and *R. rhipicephalii* str EH, suggesting efficient simultaneous transstadial transmission. The third tick was not infected with either *Rickettsia* species. The full engorgement rate, engorged nymph weight, molting rate, and adult tick weight were similar between the three progeny groups (Table S2 in Supplemental File 1). Unfortunately,

evaluation of transovarial transmission could not be performed, because none of the adult ticks from these three groups became fully engorged and therefore could not lay eggs.

**R. hoogstraalii str CS exhibits growth characteristics distinct from *R. rhipicephalii* str EH *in vitro*.** Both *R. hoogstraalii* str CS and *R. rhipicephalii* str EH were replicated in four different mammalian and tick cell lines, including Vero 81, HUVEC (ATCC CRL-1730), BGMK (ATCC PTA-4594), and the *Ixodes scapularis* cell line (IDE8) (19). Both bacteria had small purple-colored coccobacillus morphology with Giemsa staining (Fig. S5 in Supplemental File 1). In Vero 81 cells examined by transmission electron microscopy, *R. hoogstraalii* str CS were scattered in the cytoplasm and appeared as coccoid or bacillary bacteria usually measuring (1.3 $\pm$ 0.2) $\mu$m $\times$ (0.35 $\pm$ 0.04) $\mu$m, while *R. rhipicephalii* str EH were grouped in the cytosol, but not enclosed in a vacuole, and measured 1.0 $\pm$ 0.13 $\mu$m $\times$ 0.31 $\pm$ 0.03 $\mu$m (Fig. 2A). Both Vero 81 and IDE8 cells were infected with a seed culture containing 1.0 $\times$ 10$^5$ *R. hoogstraalii* str CS or *R. rhipicephalii* str EH. After incubation with the initial inoculum and washing, the infected cell layers and culture supernatants were quantified for rickettsiae. In general, *R. hoogstraalii* str CS had a higher growth rate than *R. rhipicephalii* str EH, while a continuous increase of *R. hoogstraalii* str CS and *R. rhipicephalii* str EH was seen in both cell lines resulting in 8 dpi in a maximum of 5.1 $\times$ 10$^7$ copies/$\mu$L and 7.2 $\times$ 10$^5$ copies/$\mu$L, respectively. In addition, *R. hoogstraalii* str CS revealed a slightly greater intracellular infection level in Vero 81 than in IDE8 (Fig. 2B). We observed that *R. hoogstraalii* str CS could induce cytopathic effect (CPE) evident as detached and dying cells undergoing lysis in Vero 81, IDE8 and HUVEC cells, although CPE was delayed in HUVEC cells (Fig. 2C; Fig. S6 in Supplemental File 1). In general, detachment and lysis of *R. rhipicephalii* str EH-infected cells were not obvious (Fig. 2C; Fig. S6 in Supplemental File 1). Furthermore, *R. hoogstraalii* str CS had more obvious plaque formation than *R. rhipicephalii* str EH, while the latter only formed a small number of plaques with high initial inocula ($>$2 $\times$ 10$^3$ copies/$\mu$L) (Fig. 2D).

**Characterization of the growth kinetics of *R. hoogstraalii* str CS and *R. rhipicephalii* str EH when cocultured in a competition assay *in vitro*.** The above results prompted us to directly compare the growth kinetics of *R. hoogstraalii* str CS and *R. rhipicephalii* str EH when cocultured. Competition assays are used for many microbial fitness studies due to the major advantage of internal control (20). We infected three cell lines, Vero 81, IDE8, and HUVEC, with different inoculum ratios (100:1, 10:1, 1:1, 1:10, and 1:100) of the two rickettsiae, and quantified the relative amounts of *R. hoogstraalii* str CS and *R. rhipicephalii* str EH on different days postinfection by specific quantitative PCR (qPCR). This qPCR was validated to reliably quantify the relative amounts of two rickettsiae in a mixed specimen (Fig. S7 in Supplemental File 1). We observed that *R. hoogstraalii* str CS was more competitive than *R. rhipicephalii* str EH in all three cell lines, especially in Vero 81 cells (Fig. 3). With 100:1 and 10:1 *R. hoogstraalii* str CS: *R. rhipicephalii* str EH input ratios, over 98% of bacteria were *R. hoogstraalii* str CS from 2 to 7 dpi. When they were cocultured with an equal initial amount (with an input ratio of 1:1), the proportion of *R. hoogstraalii* str CS increased from 50% to over 80% by 1 dpi. When the input ratio was 1:10, *R. hoogstraalii* str CS also accounted for over 50% of bacteria during 2 to 7 dpi in Vero 81 and HUVEC cells. Only when the input ratio of *R. hoogstraalii* str CS to *R. rhipicephalii* str EH was decreased to 1:100, the growth of *R. hoogstraalii* str CS was inhibited by *R. rhipicephalii* str EH, especially in IDE8 cells (Fig. 3A). We also calculated the growth curves to determine if different growth kinetics were due to loss in infectivity. We found both rickettsiae had continuous replication in all three cell lines (Fig. 3B).

**The actin-based-motility of *R. hoogstraalii* str CS and *R. rhipicephalii* str EH *in vitro*.** The exploitation of the host cell actin cytoskeleton is one of the important mechanisms for rickettsiae to promote motility and cell-to-cell spread. In addition, rickettsiae polymerize tails consisting of unbranched actin filaments differently from other bacteria, such as *Listeria monocytogenes* and *Shigella flexneri* (21). We thus investigated the morphologic appearance of actin tails at various times after infection with *R. hoogstraalii* str CS and *R. rhipicephalii* str EH in HUVEC cells. Double fluorescence staining of

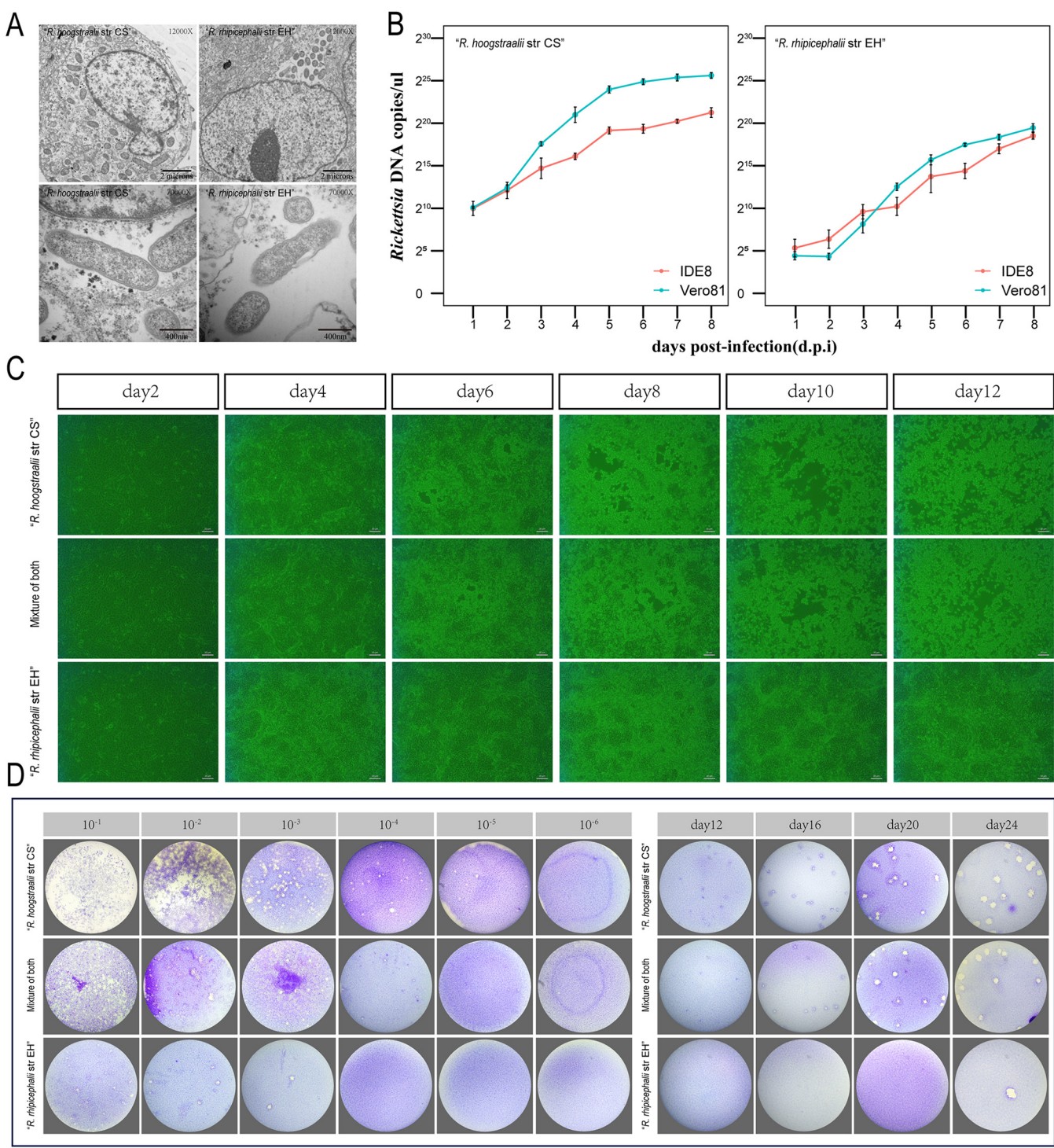

**FIG 2** Comparison of growth characteristics between *Rickettsia hoogstraalii* str CS and *Rickettsia rhipicephalii* str EH. (A) Transmission electron micrographs of Vero 81 cells infected with *R. hoogstraalii* str CS and *R. rhipicephalii* str EH. Photomicrographs were captured with an H7650 transmission electron microscope camera. (B) Growth curves of *R. hoogstraalii* str CS and *R. rhipicephalii* str EH in Vero 81 cells and IDE8 tick cells over 196 h. Error bars represent the standard deviation of the mean. (C) Cytopathic effect in Vero 81 cells induced by *R. hoogstraalii* str CS, *R. rhipicephalii* str EH, or a 1:1 mixture of both species (scale bar = 20 $\mu$m). (D) Plaque formation in Vero 81 cells by *R. hoogstraalii* str CS, *R. rhipicephalii* str EH, or a 1:1 mixture of both species at multiple MOIs and times. (i) Plaque formation with multiple MOIs at 13 days postinfection (dpi). (ii) Plaque formation after inoculation with 2 × 10³ copies/$\mu$L at different dpi.

*R. hoogstraalii* str CS-infected cells demonstrated that actin tails were frequently present and predominantly long and unbranched at both 2 and 4 dpi (Fig. 4A and B). However, despite several attempts, we failed to observe actin tail formation at the pole of *R. rhipicephalii* str EH at 2 dpi. In most instances, no actin tails were observed at 4

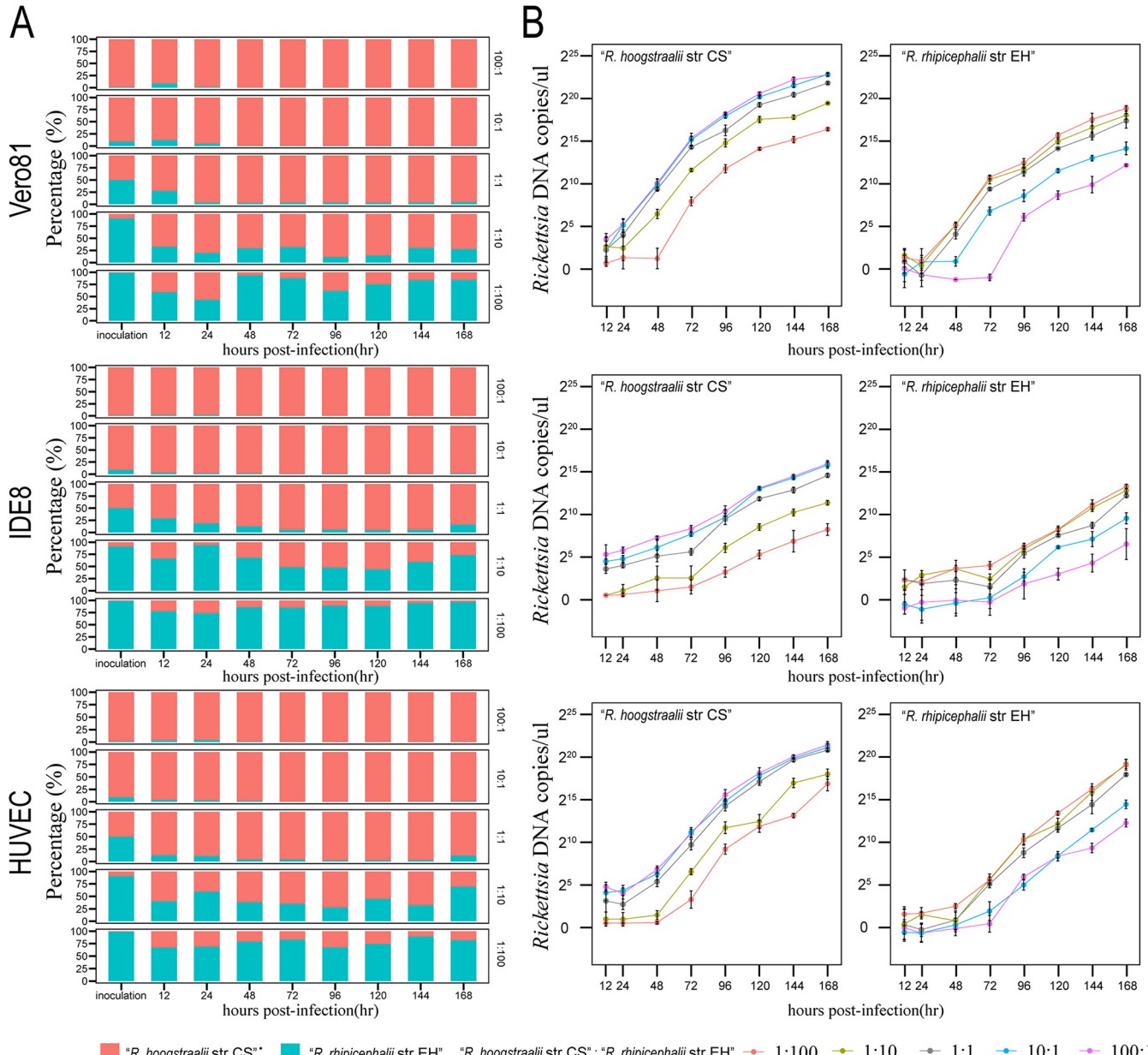

**FIG 3** Comparison of competitiveness between *Rickettsia hoogstraalii* str CS and *Rickettsia rhipicephalii* str EH when cocultured. (A) Proportions of DNA of *R. hoogstraalii* str CS (red) and *R. rhipicephalii* str EH (blue) based on quantitative PCR (qPCR) at ratios of 100:1, 10:1, 1:1, 1:10, and 1:100 *R. hoogstraalii* str CS: *R. rhipicephalii* str EH (N = 3 per group). (B) Growth curve based on qPCR at ratios of 1:100 (red), 1:10 (dark yellow), 1:1 (gray), 10:1 (blue), and 100:1 (violet) *R. hoogstraalii* str CS:*R. rhipicephalii* str EH (N = 3 per group). Error bars represent the standard deviation of the mean.

dpi (Fig. 4D), while only at occasional confocal scanning an actin tail was seen (Fig. S8A in Supplemental File 1). In addition, we observed *R. hoogstraalii* str CS diffusely scattered in multiple cells but *R. rhipicephalii* str EH confined to only a small number of cells (Fig. 4A and B). We also performed the same assays using Vero 81 cells and observed similar actin tails with *R. hoogstraalii* str CS. However, we did not find any actin tails with *R. rhipicephalii* str EH in the Vero 81 cells (Fig. S8B in Supplemental File 1).

We also compared genes encoding proteins related to the capacity of rickettsiae to promote directional actin polymerization, including RickA (22) and Sca2 (23). We did not observe any amino acid insertions or deletions in RickA and Sca2 that would result in reorganization or disruption of key domains such as the WASP (Wiskott-Aldrich syndrome protein) homology 2 (WH2) domain of RickA (Fig. S9 in Supplemental File 1).

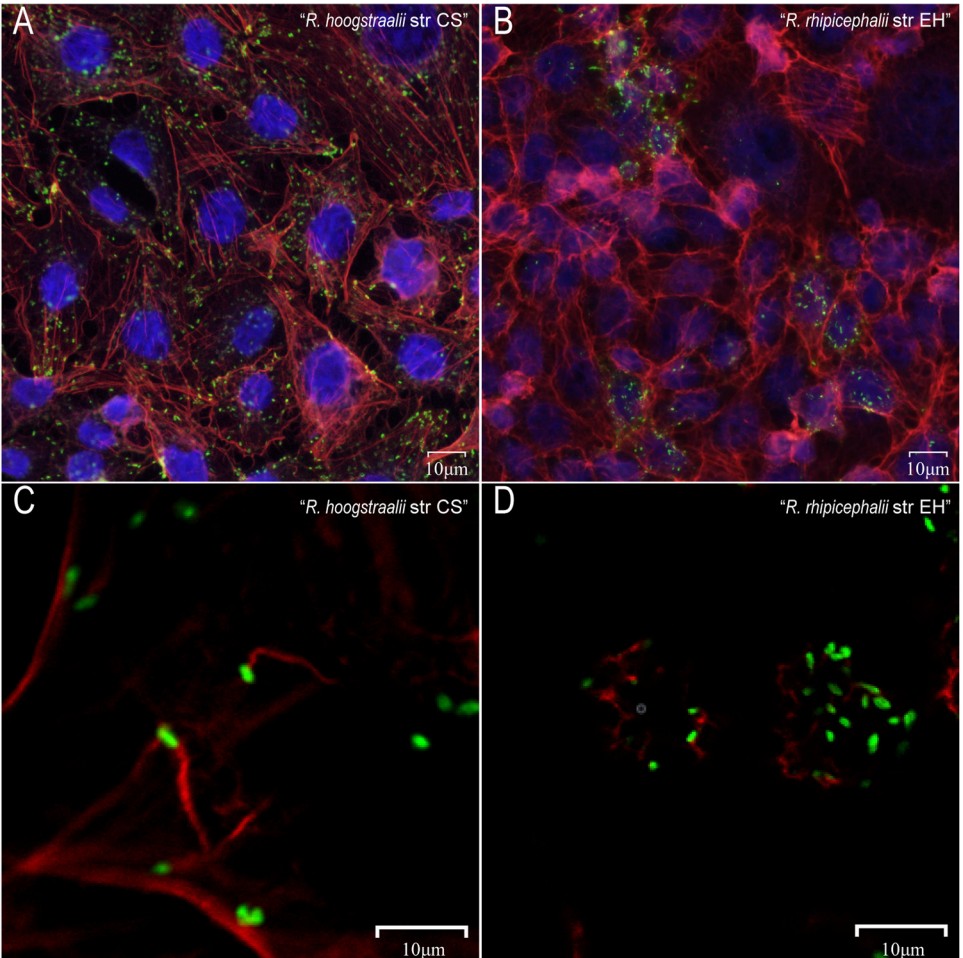

**FIG 4** Comparison of Actin Polymerization between *Rickettsia hoogstraalii* str CS and *Rickettsia rhipicephalii* str EH. HUVEC cells were infected with *R. hoogstraalii* str CS and *R. rhipicephalii* str EH. After 2 and 4 d postinfection (dpi), the cultures were fixed and stained for rickettsiae (green) and actin (red). (A) *R. hoogstraalii* str CS were diffusely scattered. (B) *R. hoogstraalii* str CS actin tails were predominantly long and unbranched at 2 dpi. (C) *R. rhipicephalii* str EH were focally concentrated. (D) *R. rhipicephalii* str EH actin tails were not obvious at 2 dpi.

## DISCUSSION

Hypothetically, simultaneous infection in ticks with two rickettsiae could facilitate or antagonize the fitness of one or the other and could be essential to understand tick-rickettsiae interaction. However, few instances of *Rickettsia* spp. coinfection in ticks has been reported. Traditional tick surveys do not efficiently detect coinfection in a single tick specimen because universal primers generally used will primarily amplify the dominant SFGR species thereby attenuating the ability to identify other species present at low levels or because amplificated bands are never sequenced to show heterogeneity. In this work, we not only identified coinfection with two *Rickettsia* species in *H. montgomeryi*, but also subsequently obtained pure isolates, i.e., *R. hoogstraalii* str CS and *R. rhipicephalii* str EH through plaque selection. Subsequent use of specific amplification primers identified 25.6% of individual ticks infected by the two rickettsiae. The work here reports the largest number to date of ticks coinfected with two rickettsiae (50/195) and the first simultaneous isolation of two bacteria that confirmed their co-occurrence. We also observed that in cell culture infection models *R. hoogstraalii* str CS overwhelmed *R. rhipicephalii* str EH with regard to CPE, plaque formation, and cellular growth when cocultured, and two *Rickettsia* species seemed to polymerize actin tails differently *in vitro*. However, *Rickettsia* phenotypes *in vitro* often do not reflect behavior

in ticks or animals, thus, further evaluation and comparison of *R. hoogstraalii* str CS and *R. rhipicephalii* str EH *in vivo* are important.

The hypothesis of transovarial interference which precluded the enzootic maintenance of two related rickettsiae within the same site has been broadly misinterpreted as two rickettsiae cannot occupy the same individual tick. We did not evaluate transovarial transmission and the tissue tropisms as in a previous test done on two *Wolbachia* novel strains coinfecting cat fleas (24), thus it is difficult to access the coinfection process. However, we enlighten the dialogue on the occurrence and importance of coinfections of two or three rickettsiae in nature. The current development of *Wolbachia*-based interference on malaria parasite *Plasmodium Falciparum* or arboviruses in mosquitos gives a promising biocontrol example (25, 26). We believe in-depth unveiling of tick-rickettsiae interactions to favor one *Rickettsia* but restrict another transmission in ticks is promising for a new attempt to biocontrol of pathogenic rickettsiae.

The mechanisms by which the infection with a first *Rickettsia* species may reduce transovarial transmission of a second *Rickettsia* remain unexplored. A previous study hypothesizes that *Rickettsia* contact-dependent growth inhibition (CDI)-like/recombination hot spot (Rhs)-like C-terminal toxin (CRCT) and CDI-like/Rhs-like C-terminal toxin antidotes (CRCA) modules circulate in the *Rickettsia* mobile gene pool, arming rickettsiae for battle over arthropod colonization (13). Another research provides evidence that the endosymbiont of *I. scapularis*, *R. buchneri*, exerts an inhibitory effect on the growth of pathogenic tick-borne bacteria in cell culture and possesses two gene cluster encoding putative antibiotic biosynthesis machinery (12). In addition, a recent report on *Wolbachia* species that infect cat fleas showed that laboratory-reared cat clones were predominantly infected by two divergent species, however, wild cat flea populations mostly harbored one species alone, suggesting what is observed in the laboratory is often not what occurs in nature (24).

We hypothesize that genetic divergence may be the driver for coinfection of *R. hoogstraalii* str CS and *R. rhipicephalii* str EH, that is these two rickettsiae may be too divergent for tick immune responses to effectively respond to superinfection. On the other hand, when transstadial transmission and co-feeding transmission happen, coinfection in the same tick is also logical. It should be noted one or more of the rickettsiae in the offspring of the engorged tick may have been obtained during blood feeding by the female tick. When individual *R. hoogstraalii* str CS- and *R. rhipicephalii* str EH-infected larvae or nymphs feed together on one vertebrate host, bacteria may transfer from one tick to another through co-feeding, without the need for the host to develop a rickettsaemia. Engorged, coinfected larvae or nymphs could then molt to nymphs or adults while carrying two rickettsiae through efficient transstadial transmission. However, transmission via co-feeding was found to be inefficient for *R. rickettsii* in *Amblyomma aureolatum* ticks (27), and a more likely route of infection would be via a rickettsemic host. Further studies are needed to determine the susceptibility of natural hosts of *H. montgomeryi* to infection with *R. hoogstraalii* str CS and *R. rhipicephalii* str EH.

Prior literature reports include *Rickettsia parkeri* and *Candidatus Rickettsia andeanae* coinfected *Amblyomma maculatum* (28, 29), *Rickettsia bellii*, *Rickettsia montanensis*, and *R. rickettsii* in *Dermacentor variabilis* (15), and *R. rhipicephali* and *R. bellii* in *Dermacentor* ticks (14). Although the coinfection reported here was in *H. montgomeryi* ticks, the question of whether such a high frequency of coinfection is common for other tick species deserves improved surveillance, perhaps by using specific primers for amplification or other methods.

Moreover, the molecular mechanisms that contribute to superinfection, rickettsial colonization, dissemination, and maintenance within tick vectors are unclear. Our *in vitro* experiment observed that *R. hoogstraalii* str CS assembled long and unbranched actin filaments but *R. rhipicephalii* str EH usually did not polymerize similar actin tails. Many intracellular bacterial pathogens that reside in the eukaryotic cell cytosol evolved mechanisms to spread from cell to cell while remaining within cells and enabling

access to cytosolic nutrients and evasion of the immune response (30). Around 30 years ago, actin-based polymerization was reported as the driving force for intracellular movement of some *Rickettsia* species and this correlates with CPE of different rickettsial species (31); recently, this *Rickettsia* dissemination mechanism was reported to be active but expendable in the tick vector as well (32). In addition, rickettsial ABM occurs in distinct phases mediated by different actin nucleators (33). For example, RickA, an activator of the host Arp2/3 complex, was initially proposed to drive motility (21), and Sca2, a mimic of host formins, was later shown to be required for motility (34). The different actin filaments assembled by *Rickettsia helvetica* and *Rickettsia peacockii* were associated with disrupted or truncated genes of Sca2, Sca4, or RickA by insertion or deletion of a region of a coding sequence, respectively (35, 36). As the ticks and animal models to evaluate *in vivo* interactions between *R. hoogstraalii* str CS and *R. rhipicephalii* str EH become available, our *in vitro* experiments of their ABM would possibly provide more insight on superinfection mechanisms.

This report has two limitations. First, insufficient numbers of laboratory ticks were used to evaluate the transovarial transmission of *R. hoogstraalii* str CS and *R. rhipicephalii* str EH as a coinfection. Only seven coinfected adults were used to feed on the rabbits, but all failed to fully engorge. Second, we have not discerned whether *R. hoogstraalii* str CS and *R. rhipicephalii* str EH coinfect the same cell or infect individual cells within a mixed culture *in vitro*.

In conclusion, these findings prove the coinfection of two rickettsiae in individual ticks. We believe the data here support a more in-depth investigation of factors that influence competition and tick-rickettsiae interactions to explore new approaches to restrict pathogenic *Rickettsia* transmission in ticks.

## MATERIALS AND METHODS

**Ethics statement.** All applicable national and/or institutional guidelines for the care and use of animals were followed. All procedures involving laboratory animals were approved by the Experimental Animal Welfare Committee of the State Key Laboratory of Pathogen and Biosecurity, Beijing Institute of Microbiology and Epidemiology under project no. 81773492.

**Initial isolation of *Rickettsia* from ticks.** A pool of 12 nymphal *H. montgomeryi* ticks derived from the progeny of a single adult female tick collected in 2018 from Dali City, Yunnan Province, southwestern China was used for isolation attempts. The pooled nymphs were immersed in 0.1% bleach and 75% ethanol and washed with sterile PBS. After manual homogenization in 1 mL Dulbecco's Modified Eagle's Medium (DMEM) with 2% fetal bovine serum (FBS), 100 $\mu$L per well of the tick homogenate was inoculated onto Vero 81 (ATCC catalog no. CCL-81) cells in 24-well culture plates, followed by incubation at 32°C for 2h. Then 1 mL per well of DMEM supplemented with 2% FBS, 200 units/mL penicillin, and 200 $\mu$g/mL streptomycin were added to the cultures. The remaining tick homogenate was used for DNA extraction. Giemsa staining and specific PCRs targeting *ompA* and *gltA* genes were used for assessing the isolation of *Rickettsia* species every week (37).

**Plaque selection of *R. hoogstraalii* str CS and *R. rhipicephalii* str EH.** Vero 81 cells were trypsinized and resuspended at $2 \times 10^5$ cells per mL in DMEM containing 10% FBS and plated at 2.5 mL per well in six-well plates. The cells were incubated at 37°C in a humidified 5% $CO_2$ atmosphere for 24 h. Aliquots of 400 $\mu$L per well of 10-fold serial dilutions of rickettsial suspensions were added and then rinsed twice with PBS after inoculation. Each well was then overlaid with 2 mL of DMEM containing 5% FBS and 0.8% agarose and, after solidification, covered with 1 mL DMEM containing 5% FBS. The plates were incubated at 32°C in a humidified 5% $CO_2$ atmosphere. Morphologically variant plaques were marked, aspirated by pipette, and mixed with 200 $\mu$L DMEM to inoculate Vero cells in 24-well culture plates for clonal expansion.

**Transmission electron microscopy of two *Rickettsia* species.** Resuspended cells from the two rickettsiae-infected Vero 81 cell cultures were centrifuged at 800 × *g* for 5 min; the supernatant was discarded, and the cell pellet was fixed in 2.5% glutaraldehyde (wt/vol) for 2h. The cells were then dehydrated with a graded series of ethanol at 50%, 70%, 90%, and 100% before being embedded in resin. The fresh resin was used to embed pellets in molds and cured for 48 h at 60°C. Ultrathin serial sections (50 to 100 nm) were cut and collected on Formvar-coated copper grids. Grids were poststained with 2% uranyl acetate for 15 min and lead citrate for 10 min. After washing with double-distilled water and drying on copper grids, pellets were viewed at 80KV using an H7650 transmission electron microscope (Hitachi, Japan).

**Comparison of growth between *R. hoogstraalii* str CS and *R. rhipicephalii* str EH *in vitro*.** Two cell lines were used. Vero 81 was maintained as above and the *Ixodes scapularis* tick cell line IDE8 was maintained in L-15B medium supplemented with 10% tryptose phosphate broth, 10% FBS, and 0.1% bovine lipoprotein (19). *R. hoogstraalii* str CS and *R. rhipicephalii* str EH were inoculated onto monolayers of Vero 81 and IDE8 cells in 24-well plates at a multiplicity of infection (MOI) of 0.1 bacteria/cell and incubated

at 32°C for 2h. Following inoculation, the cells were washed, fresh medium was added, and the plates were incubated at 32°C for up to 192 h. At the designated time points, the monolayers were resuspended by scraping. DNA was extracted from 200 $\mu$L aliquots of the cell suspensions using a MiniBEST Viral RNA/DNA Extraction kit (TaKaRa, Japan) according to the manufacturer's instructions. A primer pair (forward, TAACTTAACAGGCAGCATA; reverse, ATTAGCCGCAGTCCCTAC) capable of amplifying the *ompA* gene (100 bp) of both *R. hoogstraalii* str CS and *R. rhipicephalii* str EH was used for qPCR with an annealing temperature of 60°C (Table S1 in Supplemental File 1). The copy numbers of each *Rickettsia* species were calculated by a standard curve method using a plasmid containing the corresponding segment.

**Comparison of CPE between *R. hoogstraalii* str CS and *R. rhipicephalii* str EH.** *Rickettsia* species infections in cultures of Vero 81, HUVEC (ATCC catalog no. CRL-1730), and BGMK (ATCC PTA-4594) maintained in DMEM with 10% FBS, and IDE8 cells were performed as described above. Cells were inoculated with *R. hoogstraalii* str CS, *R. rhipicephalii* str EH, or a mixture at the same initial MOI of 0.05. Giemsa-stained preparations were examined and the CPE of the rickettsiae in the cells was observed on different days postinoculation under a light microscope (Zeiss, Germany). Images were acquired and processed using ZEN lite 2012 software.

**Comparison of plaques between *R. hoogstraalii* str CS and *R. rhipicephalii* str EH.** The plaque assays were performed in Vero 81 cells as described above. Briefly, plates were incubated at 32°C in a 5% $CO_2$ atmosphere to observe the formation of plaques after inoculation with multiple MOIs at designated time points. The monolayers were fixed with saline formalin (3.7% to 4.0% formaldehyde) and stained for 60 min with 1% crystal violet. The monolayers were washed several times with water to remove agarose and air-dried before plaque counting.

**Competition assay between *R. hoogstraalii* str CS and *R. rhipicephalii* str EH *in vitro*.** To evaluate rickettsial replication in a competition assay, *R. hoogstraalii* str CS and *R. rhipicephalii* str EH were mixed at different ratios and inoculated onto Vero 81, HUVEC, and IDE8 cell monolayers grown in 24-well plates. At designated time points, the monolayers were frozen at −80°C for 1h and harvested. DNA extraction was performed as described above. The following primers, targeting the *ompA* gene, were used for differentiation between *R. hoogstraalii* str CS and *R. rhipicephalii* str EH genomes in competition experiments: YN1ompA122: 5′-CATCGTCATCACCGTCTA-3′ (forward) and YN1ompA512: 5′-GCTAAT GGTAATCCTGCT-3′ (reverse) for *R. hoogstraalii* str CS; YN2-OmpA115: 5′-GTTATTATACCTCCTCCATC-3′ (forward) and YN2-OmpA323: 5′-TTGCCTGTTACTATTACTGC-3′ (reverse) for *R. rhipicephalii* str EH. For detecting total rickettsiae, a primer pair (forward-TAACTTAACAGGCAGCATA, reverse-ATTAGCC GCAGTCCCTAC), capable of amplifying the *ompA* gene (100 bp) of both *R. hoogstraalii* str CS and *R. rhipicephalii* str EH was used (Table S1 in Supplemental File 1). A primer annealing temperature of 60°C was used for all assays. Nine-point standard curves ($1 \times 10^1$ copies/$\mu$L to $1 \times 10^9$ copies/$\mu$L) were utilized to quantify the bacterial load.

**Comparison of actin polymerization between *R. hoogstraalii* str CS and *R. rhipicephalii* str EH.** HUVEC and Vero 81 cells cultured on 14 mm coverslips in 24-well plates were infected with *R. hoogstraalii* str CS and *R. rhipicephalii* str EH and then incubated at 32°C for 2h. Fresh medium containing 0.5% FBS was added to each well. The infections were allowed to progress for 48h or 96h, and fixed with 3.7% paraformaldehyde for 15 min. The cells were permeabilized with 0.1% Triton X-100 diluted with PBS for 5 min and immersed in 5% bovine serum albumin in PBS for 1h to allow binding with nonspecific sites. The rickettsiae were labeled with *R. hoogstraalii* str CS or *R. rhipicephalii* str EH polyclonal antibody (prepared by intraperitoneal injection of each isolated *Rickettsia* into BALB/c mice) and then with anti-mouse immunoglobulin-Alexa Fluor 488 secondary antibody (Invitrogen) for 30 min at 37°C. F-actin was labeled with Alexa Fluor 568-phalloidin (Invitrogen) for 45 min. All steps were followed by three washes in PBS. Coverslips were sealed with an antifade mounting medium (Invitrogen). Images were acquired on an Olympus microscope with a 60 1.4-numerical-aperture (NA) oil immersion objective (Olympus).

**Comparison of genomes between *R. hoogstraalii* str CS and *R. rhipicephalii* str EH.** Vero 81 cell cultures infected with the two *Rickettsia* species in T75 flasks were harvested by scraping into 5 mL fresh DMEM and *Rickettsia* were released from the cells by repeated passages through a 27G needle and then semipurified by centrifugation first at 800 × *g* for 5 min to remove any remaining intact Vero 81 cells and cell debris and then at 17,000 × *g* for 10 min to pellet the bacteria in the supernatant. DNA was extracted from the semipurified bacteria using a High Pure PCR Template Preparation kit (Roche, Germany). RNA was also extracted from the bacteria to annotate the genomes of *Rickettsia* spp., using an RNeasy Minikit (Qiagen, USA).

Libraries were sequenced on the MiSeq platform with a paired-end (PE) 300 bp sequencing strategy. Raw reads were filtered using AfterQC v0.9.6 with default parameters. High-quality reads were aligned to the *Chlorocebus sabaeus* (African green monkey) genome (GenBank assembly accession no. GCA_000409795.2) using e bowtie2 v2.4.1 with default parameters, to remove the genome sequence of the host Vero cells (the Vero cell line was initiated from the kidney of an African green monkey). Both members of a read pair were discarded if one read matched the *C. sabaeus* genome by using samtools v1.9 with parameters -f 4. rickettsiae genomes were assembled using SPAdes v3.15.2 with parameters -isolate. Busco v4.1.2 was used to evaluate the completeness of the assembly with parameters -l rickettsiales_odb10. The OrthoANI values between genomes were calculated by the OrthoANI v0.91. Gene annotation was accomplished by Prokka v1.14.6 with default parameters. GC content and GC skew value were calculated by GCcalc v1.0.0. KEGG annotation was accomplished by BlastKOALA at the KEGG website (http://www.kegg.jp/blastkoala/). RNA and DNA sequencing data coverage were summarized by bedtools v2.30.0 with parameters coverage -mean. Genes involved in metabolism and human diseases were selected by KOs below A09100 and A09160. TPM (transcripts per kilobase of exon model per million mapped reads of genes) were calculated by stringtie v1.3.4d after RNA sequencing data were mapped onto the genome by bowtie2.

**Reconstruction of the phylogenomic tree of *R. hoogstraalii* str CS and *R. rhipicephalii* str EH.** First, Prokka v1.14.6 was used to annotate the genome. Gene families were identified using orthofinder v2.5.4 with default parameters among 25 *Rickettsia* species and two outgroup species. Single-copy gene families (*n* = 332) were used for subsequent phylogenetic analysis. The protein sequences of these single-copy genes were aligned using MUSCLE v3.8.1551. Gblocks v0.91b was used to select conserved blocks from aligned sequences. Finally, Raxml v8.2.12 was used to generate a phylogenomic tree and count branch support.

**Mixed sample genome assembly and genome distinguished.** Raw reads from the mixed sample were applied to quality control and host filter pipeline as described above. Rickettsiae genomes were assembled using SPAdes v3.15.2 with parameters –meta. Assembled sequences were aligned to *Rickettsia massiliae* and *Rickettsia hoogstraalii* genome by NUCmer v3.1. Sequences were split into two genomes based on overall sequence identity. The phylogenomic tree was reconstructed as described above with 317 single-copy gene families used.

**Data availability.** The whole genomes of *Rickettsia hoogstraalii* str CS and *Rickettsia rhipicephalii* str EH were submitted to China National Center for Bioinformation (CNCB) (https://www.cncb.ac.cn/) with accession PRJCA008681 and NCBI (https://www.ncbi.nlm.nih.gov) with BioProject PRJNA882799 for *Rickettsia hoogstraalii* strain CS and PRJNA882798 for *Rickettsia rhipicephali* strain EH.

## SUPPLEMENTAL MATERIAL

Supplemental material is available online only.

**SUPPLEMENTAL FILE 1**, PDF file, 1.2 MB.

## ACKNOWLEDGMENTS

We would like to express our great appreciation to J. Stephen Dumler (Department of Pathology, Uniformed Services University) for his insightful suggestions on the work, critical review, and careful editing of the manuscript. We thank Ulrike Munderloh (University of Minnesota) and the Tick Cell Biobank (University of Liverpool) for the IDE8 cell line.

This study was supported by the Natural Science Foundation of China (grant no. 81621005 and 81773492), the State Key Research Development Program of China (no. 2019HK125), and the Natural Science Foundation of Shandong Province, China (no. ZR2020QH299). L.B.-S. was supported by the United Kingdom Biotechnology and Biological Sciences Research Council (no. BB/P024270/1).

N.J. W.-C.C., designed and supervised research. C.-H.D., X.-M.C., W.-C.C., N.J., L.-Y.X., Y.D., Z.-H.H., and M.-G.Y., collected samples. Y.-S.P., L.-Y.X., M.-Z.Z., D.-Y.Z., W.W., Y.S., Q.-Y.L., Q.W., and L.-J.L., performed experiments. Y.-S.P., L.-F.D., N.J., W.-C.C., L.Z., and R.-Z.Y., analyzed the data. L.B.-S. provided materials and edited the manuscript, and N.J. and W.-C.C. wrote the paper.

We declare no competing interests.

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
