## [Reviewer comments · Microbiology Spectrum]

Microbiology Spectrum

Co-infection of Two Rickettsia Species in a Single Tick Species Provides New Insight into Rickettsia-Rickettsia and Rickettsia-vector Interactions

na jia, Yu-Sheng Pan, Xiao Cui, Li-Feng Du, Luo-Yuan Xia, Chun-Hong Du, Lesley Bell-Sakyi, Ming-Zhu Zhang, Dai-Yun Zhu, Yi Dong, Wei Wei, Lin Zhao, Yi Sun, Qingyu Lv, Run-Ze Ye, Zhi-Hai He, Qian Wang, Liang-Jing Li, Ming-Guo Yao, Wu-Chun Cao, tao xiong, and Jia-Fu Jiang

Corresponding Author(s): na jia, Beijing Institute of Microbiology and Epidemiology

Review Timeline:

Submission Date:

August 26, 2022

Accepted:

September 15, 2022

Editor: Jeffrey Gralnick

Reviewer(s): The reviewers have opted to remain anonymous.

Transaction Report:

DOI: <https://doi.org/10.1128/spectrum.02323-22>

September 15, 2022

Dr. na jia
Beijing Institute of Microbiology and Epidemiology
Beijing
China

Re: Spectrum02323-22 (Co-infection of Two Rickettsia Species in a Single Tick Species Provides New Insight into Rickettsia-Rickettsia and Rickettsia-vector Interactions)

Dear Dr. Na Jia:

Based on your revisions to the prior round of review, your manuscript has been accepted, and I am forwarding it to the ASM Journals Department for publication. You will be notified when your proofs are ready to be viewed.

Sincerely,

Jeffrey Gralnick
Editor, Microbiology Spectrum
